# Decatungstate-catalyzed radical disulfuration through direct C-H functionalization for the preparation of unsymmetrical disulfides

Jingjing Zhang[1] & Armido Studer [1✉]

Unsymmetrical disulfides are widely found in the areas of food chemistry, pharmaceutical industry, chemical biology and polymer science. Due the importance of such disulfides in various fields, general methods for the nondirected intermolecular disulfuration of C-H bonds are highly desirable. In this work, the conversion of aliphatic $C(sp^3)$-H bonds and aldehydic $C(sp^2)$-H bonds into the corresponding C-SS bonds with tetrasulfides (RSSSSR) as radical disulfuration reagents is reported. The decatungstate anion ($[W_{10}O_{32}]^{4-}$) as photocatalyst is used for C-radical generation via intermolecular hydrogen atom transfer in combination with cheap sodium persulfate ($Na_2S_2O_8$) as oxidant. Herein a series of valuable acyl alkyl disulfides, important precursors for the generation of RSS-anions, and unsymmetrical dialkyl disulfides are synthesized using this direct approach. To demonstrate the potential of the method for late-stage functionalization, approved drugs and natural products were successfully C-H functionalized.

[1] Organisch-Chemisches Institut, Westfälische Wilhelms-Universität, Corrensstrasse 40, 48149 Münster, Germany. ✉email: studer@uni-muenster.de

Owing to the unique properties of disulfides in folding, stability, and biological functions of peptides and proteins, the disulfide moiety is of high importance in such biomacromolecules[1–4]. Moreover, disulfides are also involved in biological systems in maintaining the cellular redox balance in cell survival[5,6]. Due to their unique pharmacological and physiochemical properties, disulfide moieties also occur in lower molecular weight biologically active natural products and in drugs[7–9], such as in members of the antifungal polycarpamine family, gombamide A, polycarpamine B and C, Z-ajoene, (-)-emethallicin E, psammaplin F and kottamide E (Fig. 1). More generally, compounds bearing a disulfide moiety are widely found in the realm of food chemistry, pharmaceutical industry, chemical biology and also in polymer science[10–12]. Consequently, the development of synthetic strategies for the preparation of disulfides, in particular unsymmetrical disulfides, has gained considerable attention in recent years[8,13–17].

However, efficient and selective synthetic methods for disulfide bond construction remain limited, especially considering the formation of unsymmetrical disulfides. Disulfides have been prepared by following either the S-S or the C-SS bond construction pathway (Fig. 2). Unsymmetrical disulfides can directly be accessed by oxidative cross coupling of two different thiols through electro-oxidation[18] or using photocatalysis[19–21]. However, these methods suffer from a low chemoselectivity providing unavoidable homo-coupling products (symmetrical disulfides). This selectivity problem was solved by using the ionic approach choosing sulfur derivatives bearing an anionic leaving group (LG) as electrophiles in combination with thiols as nucleophiles[22,23]. Along these lines, the disulfide exchange reaction represents the currently most commonly applied method for the preparation of unsymmetrical disulfides[24]. Considering C-SS bond formation, that approach has benefited from the development of electrophilic and nucleophilic "masked" disulfuration reagents[25,26]. These transformations proceed through metal-catalyzed cross-coupling[27,28] or substitution reactions[15,29–31]. It is worth mentioning that in these disulfuration processes, one reaction component carries both sulfur atoms

rendering such methods valuable for the preparation of unsymmetrical disulfides. Nevertheless, these approaches remain limited due to the tedious preparation of the requested rather complex disulfuration reagents.

Radical chemistry has become highly popular and offers alternative retrosynthetic routes for bond construction under mild conditions, and has therefore significantly influenced the reaction design in modern organic chemistry[32–38]. Recently, Pratt and co-workers have successfully prepared a diverse set of unsymmetrical disulfides via radical substitution with tetrasulfides[39] or trisulfide-1,1-dioxides[40] as C-radical acceptors, profiting from the inherent cleavage tendency of S-S bonds to generate thermodynamically stable perthiyl (RSS•) or sulfonyl radicals. The C-radicals engaging in such disulfurations were generated from O-acyl oximes, Barton PTOC esters or from carboxylic acids. Meanwhile, Wang and co-workers introduced a metal-free disulfuration of anilines via their corresponding diazonium salts[41]. In these reactions, the C-radical intermediates are generated through the transformation of a functional group which is accompanied by the generation of additional waste. It is important to note that Pratt and co-workers disclosed very recently an elegant regioselective radical C-H disulfuration that proceeds via Hofmann–Löffler–Freytag type chemistry, where C-radical generation is achieved by an intramolecular 1,5-HAT to an N-centered radical[40]. However, methods that allow the formation of C − SS bonds via an intermolecular HAT process are underdeveloped.

In this work, we explore a more atom-economic radical disulfuration strategy that operates without the need of a pre-functionalized radical precursor via direct C-H bond functionalization using a hydrogen atom transfer (HAT) catalyst. This route significantly expands the substrate scope for radical disulfuration, especially considering late-stage functionalization of pharmaceuticals.

## Result and discussion

**Reaction design.** In recent years, photocatalysis has been successfully applied for hydrogen-atom transfer (HAT) mediated radical functionalization of C-H bonds. Among the photocatalysts that operate via HAT, the decatungstate anion ($[W_{10}O_{32}]^{4-}$), featuring electron-deficient oxygen atoms, has been widely explored due to its ability to catalytically cleave even strong inert aliphatic C($sp^3$)-H bonds[42–52]. We therefore selected the tetrabutylammonium decatungstate 1 as the photocatalyst to realize the direct intermolecular radical C-H disulfuration. Our reaction design and mechanistic hypothesis are depicted in Fig. 3, exemplified for the functionalization of cyclohexane (2a). Irradiation of the decatungstate anion of 1 will lead to the reactive excited state 1*, which is known to abstract an H atom from an alkane such as 2a to give the cyclohexyl radical 2a' along with $H[W_{10}O_{32}]^{4-}$. Such alkyl radicals are readily trapped by a tetrasulfide to give the targeted disulfuration product 4 and the corresponding thermodynamically stable perthiyl radical (RSS•) which can dimerize ($k = 6 \times 10^9 \, M^{-1} \, s^{-1}$, for R = t-Bu) to reform the starting tetrasulfide[53]. The photocatalysis cycle would be closed by deprotonation of $H[W_{10}O_{32}]^{4-}$ to give $[W_{10}O_{32}]^{5-}$ which should then be oxidized via single-electron transfer (SET) to regenerate the starting decatungstate 1. Since the perthiyl radical (RSS•) is thermodynamically stable, hydrogen abstraction of this RSS-radical from $H[W_{10}O_{32}]^{4-}$ to regenerate 1 is not very likely. Therefore, an external stoichiometric oxidant is probably required to drive the process. Moreover, the RSSH formed in such a HAT would be a very good hydrogen atom donor that would competitively react with the alkyl radical 2a' suppressing the desired disulfuration. A challenge associated with our design is

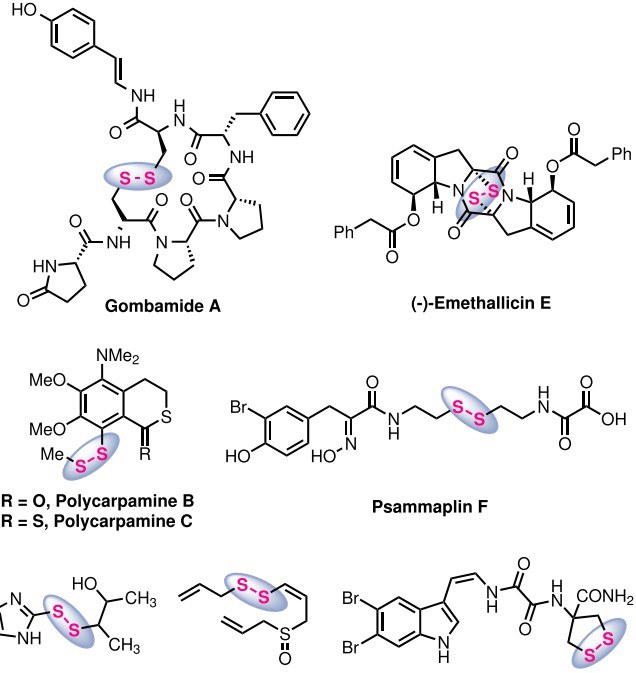

**Fig. 1 The disulfide moiety in natural products and drugs.** Unsymmetrical disulfides in natural products and drugs.

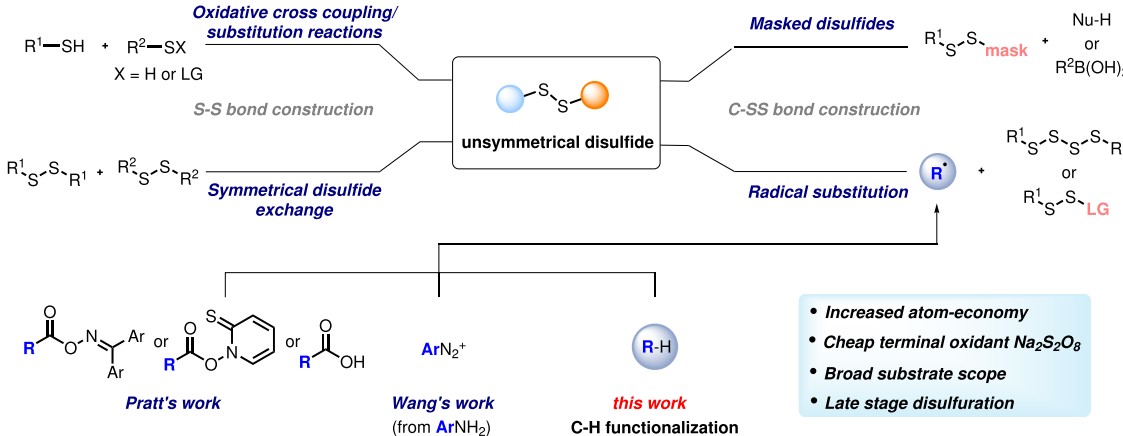

**Fig. 2 State of art methods for the preparation of unsymmetrical disulfides.** Reported methods for preparation of unsymmetrical disulfides and this work.

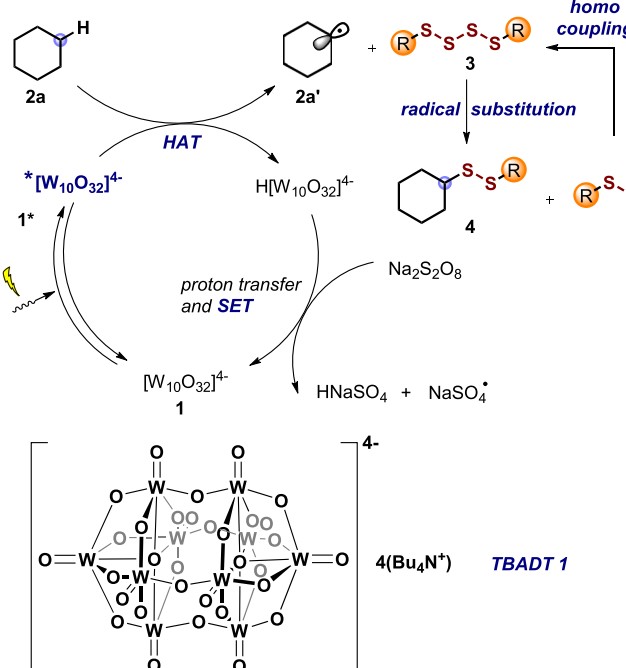

**Fig. 3 Reaction design.** The suggested mechanism for direct C-H disulfuration via decatungstate photocatalysis.

the potential unwanted direct oxidation of the tetrasulfide with the terminal oxidant. We selected cheap $Na_2S_2O_8$, that is known to oxidize $[W_{10}O_{32}]^{5-}$ [54], as the oxidant in the targeted C-H disulfuration. In this redox reaction, the $NaSO_4$-radical will be generated that may also act as an electrophilic reagent engaging in C-H abstraction from the substrate[55]. Another problem that might appear is the potential direct reduction or oxidation of the tetrasulfide by the redox catalyst which must also be circumvented.

**Investigation of the reaction conditions**. Optimization studies were performed with cyclohexane **2a** and the tetrasulfide **3a** as the limiting reagent targeting the unsymmetrical disulfide **4a** (Table 1). Careful experimentation revealed that this reaction is best conducted with TBADT **1** (2 mol%) and sodium persulfate $Na_2S_2O_8$ (1.5 equiv.) in acetonitrile/water (2:1) upon irradiation with a Kessil 40 W 390 nm lamp for 12 h at 60 °C to provide the disulfuration product **4a** in 86% isolated yield (entry 1). The reaction did not occur in the absence of the photocatalyst **1** (entry 2) and showed a large solvent dependence. In a $CH_2Cl_2$/water mixture, only traces of **4a** were formed (entry 3) and in acetonitrile a low yield was noted (entry 4). Decreasing the reaction temperature to 25 °C resulted in a lower conversion (46%, entry 5). Interestingly, without added $Na_2S_2O_8$, the disulfide **4a** could still be obtained, albeit in low 12% yield (entry 6). We currently assume that the tetrasulfide **3a** is also capable of reoxidizing the photocatalyst, though at a much lower efficiency. This catalyst reoxidation is slightly more efficient in aqueous acidic acetonitrile, where **4a** was formed in 28% yield in the absence of the terminal oxidant (entry 7). An alternative is the reaction of the dithyl radical with $H[W_{10}O_{32}]^{4-}$ to regenerate the catalyst and the dithiol. However, $tBuSSH$ was not detected in the reaction mixture. Replacing $Na_2S_2O_8$ with $Na_2S_2O_4$, the product **4a** was formed in trace amounts (entry 8). Upon replacing the tetrasulfide **3a** by a trisulfide-1,1-dioxide ($p$-TolSO$_2$SS$t$-Bu), the disulfide **4a** was observed in traces only (entry 9). When using cyclohexane **2a** as the limiting reagent (**2a** 0.3 mmol, **3a** 0.45 mmol) under otherwise identical conditions, **4a** was obtained in 29% GC yield.

**Substrate Scope**. With the optimized conditions in hand, we turned our attention to explore the substrate scope (Fig. 4). The tetrasulfides were readily obtained from the corresponding thiols upon treatment with $S_2Cl_2$ (see Supplementary Information). *Tert*-butyl (**3a**), cyclohexyl (**3b**), n-nonyl (**3c**), benzyl (**3d-3g**), heterocyclic (**3 h**), phenyl (**3i**), benzoyl (**3j**) and ester (**3 l**) substituted tetrasulfides were included into these investigations. In

### Table 1 Optimization of the reaction conditions[a].

| Entry | Deviation from standard conditions | Yield[b] |
|---|---|---|
| **1** | **standard conditions** | **86%** |
| 2 | without TBADT | trace |
| 3 | $CH_2Cl_2/H_2O$ as solvent | trace[c] |
| 4 | $CH_3CN$ as solvent | 18% |
| 5 | 25 °C | 46% |
| 6 | without $Na_2S_2O_8$ | 12% |
| 7 | without $Na_2S_2O_8$, $CH_3CN$/(HCl aq.) as solvent | 28%[d] |
| 8 | $Na_2S_2O_4$ in replace of $Na_2S_2O_8$ | trace |
| 9 | SS-(*tert*-butyl) 4-methylbenzenesulfono-(dithioperoxoate) as disulfuration reagent | trace[e] |

[a]Reaction conditions: tetrasulfide **3a** (0.3 mmol, 1.0 equiv.), cyclohexane **2a** (3 mmol, 10 equiv.), TBADT 2 mol%, $Na_2S_2O_8$ (0.45 mmol, 1.5 equiv.), solvent 3 mL ($CH_3CN/H_2O$, $v/v$, 2/1), 390 nm, Ar, 60 °C, and 12 h. [b]Isolated yield. [c]Solvent 3 mL ($CH_2Cl_2/H_2O$, $v/v$, 2/1). [d]Solvent 3 mL [$CH_3CN$/(HCl aq. 1.0 M), $v/v$, 2/1]. [e]Without $Na_2S_2O_8$.

addition, the more complex cysteine-derived tetrasulfide **3k** was also prepared. Scope with respect to the tetrasulfide component was tested first with cyclohexane (**2a**) as the C-radical precursor. All tetrasulfide reagents **3a**-**3l** performed well to provide the disulfuration products **4a**-**4k'** with moderate to high yields. Some reactions were repeated using cyclohexane as the limiting reagent; however, low yields were obtained in these cases (see **4b**-**4d**). Replacement of the *tert*-butyl substituent in the parent reagent **3a** with cyclohexyl or a long-chain alkyl group was well tolerated and the products **4b** and **4c** were obtained in high yields (89% and 81%). Bisbenzyl-substituted tetrasulfides bearing either electron-donating or electron-withdrawing *para*-substituents at the arene

**Fig. 4 Reaction scope.** Isolated yields are provided. Selectivity determined by gas chromatography is reported as the percentage of the major regioisomer. GC yields with alkanes as the limiting reagent in parentheses: substrate **2** (0.3 mmol) and tetrasulfide **3** (0.45 mmol). [a]Method A: tetrasulfide **3** (0.3 mmol, 1.0 equiv.), **2** (3 mmol, 10 equiv.), TBADT 2 mol%, Na$_2$S$_2$O$_8$ (0.45 mmol, 1.5 equiv.), solvent 3 mL (CH$_3$CN/H$_2$O, v/v, 2/1), 390 nm, Ar, 60 °C, and 12 h. [b]Method B: tetrasulfide **3** (0.3 mmol, 1.0 equiv.), **2** (3 mmol, 10 equiv.), TBADT 2 mol%, solvent 3 mL [CH$_3$CN/(HCl aq. 1.0 M), v/v, 2/1], 390 nm, Ar, 60 °C, and 12 h. [c]Using CH$_3$CN/H$_2$O (v/v, 2/1) as solvent. [d]**2** (0.3 mmol, 1.0 equiv.), tetrasulfide **3** (0.45 mmol, 1.5 equiv.), TBADT 2 mol%, solvent 3 mL [CH$_3$CN/(HCl aq. 1.0 M), v/v, 2/1], 390 nm, Ar, 60 °C, and 12 h. [e]Tetrasulfide **3** (0.3 mmol, 1.0 equiv.), substrate **2** (1.5 mmol, 5.0 equiv.), TBADT 2 mol%, Na$_2$S$_2$O$_8$ (0.45 mmol, 1.5 equiv.), solvent 3 mL (CH$_3$CN/H$_2$O, v/v, 2/1), 390 nm, Ar, 60 °C, and 12 h. [f]Determined by $^1$H NMR spectroscopy. [g]Tetrasulfide **3** (0.45 mmol, 1.5 equiv.), substrate **2** (0.3 mmol, 1.0 equiv.), TBADT 2 mol%, Na$_2$S$_2$O$_8$ (0.45 mmol, 1.5 equiv.), solvent 3 mL (CH$_3$CN/H$_2$O, v/v, 2/1), 390 nm, Ar, room temperature, and 12 h.

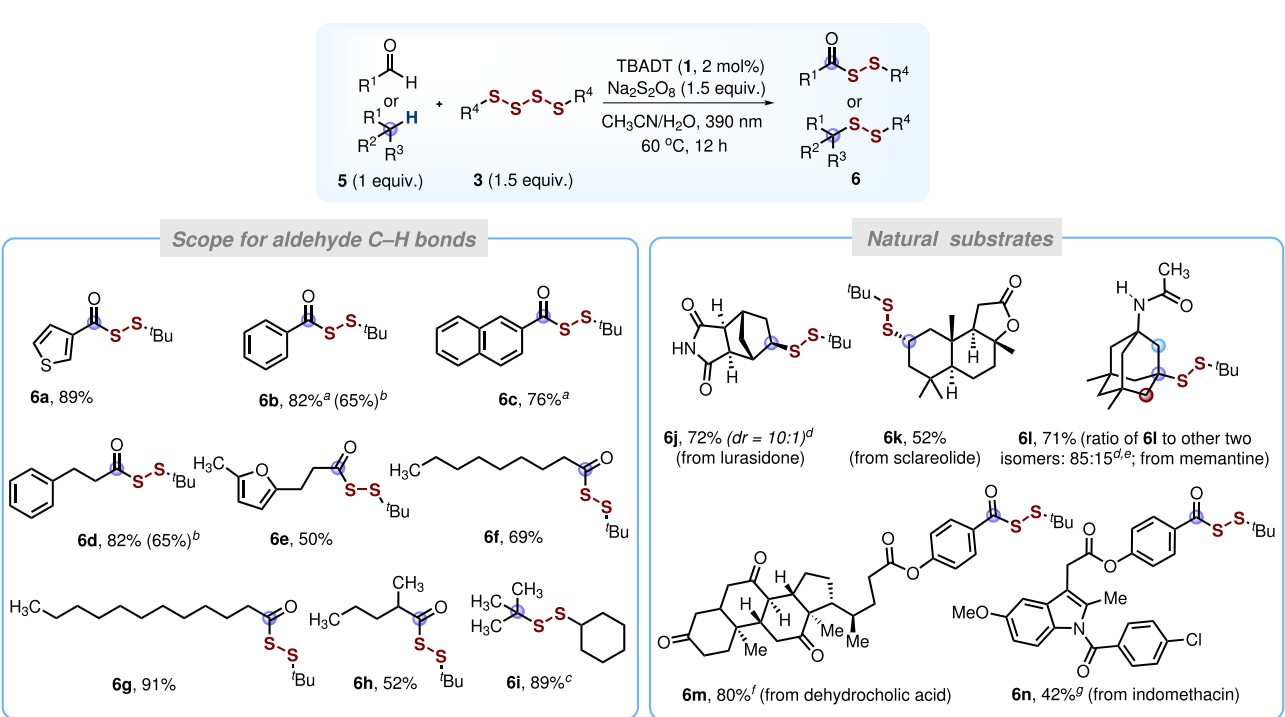

**Fig. 5 Products obtained by disulfuration of aldehydes and functionalization of natural compounds.** Isolated yields are provided. Reaction conditions: substrate **5** (0.3 mmol, 1.0 equiv.), tetrasulfide **3a** (0.45 mmol, 1.5 equiv.), TBADT 2 mol%, Na$_2$S$_2$O$_8$ (0.45 mmol, 1.5 equiv.), solvent 3 mL (CH$_3$CN/H$_2$O, v/v, 2/1), 390 nm, Ar, 60 °C, and 12 h. [a]Reaction time: 4 h. If the reaction time is extended to 12 h, some of the products will decompose into aroylmonosulfides (see SI for details). [b]Reaction conducted at room temperature. [c]With tetrasulfide **3b** (1.5 equiv.). [d]Determined by $^1$H NMR spectroscopy. [e]Reaction run in the absence of Na$_2$S$_2$O$_8$ in 3 mL [CH$_3$CN/(HCl aq. 1.0 M), v/v, 2/1]. [f]Reaction conditions: substrate **5m** (0.2 mmol, 1.0 equiv.), 5 h. [g]Reaction conditions: substrate **5n** (0.1 mmol, 1.0 equiv.), room temperature, 12 h.

moiety showed good reactivity (**4d**-**4g**, 76–83%). Moreover, the tetrasulfide **3 h** reacted smoothly to afford the 2-furyl-substituted disulfide **4 h** in 71% isolated yield. However, with the diphenyltetrasulfide **3i** the targeted **4i** was not formed. Instead the monosulfide **4i'** was obtained (72%). We found that the reaction in the absence of Na$_2$S$_2$O$_8$-oxidant gave the targeted disulfide **4i** as minor product in 26% yield besides the monosulfide **4i'** (37%). A similar outcome was noted for the bisbenzoyltetrasulfide **3j** providing under the standard conditions exclusively the monosulfide **4j'** (53%). In this case, the oxidant-free protocol afforded the monosulfide **4j'** (37%) and the targeted disulfide **4j** (41%). We found that diaryl, diacyl tetrasulfides and the corresponding targeted disulfides are not perfectly stable under the reaction conditions eventually leading to the isolated monosulfides. It is worth mentioning that acyl alkyl disulfides (see **4j** and also congeners depicted in Fig. 5) are valuable precursors for the preparation of difficult-to-access hydropersulfides (RSSH)[56–58], which are excellent hydrogen-atom donors[59] playing a regulatory role in redox processes in biology[60–62].

The scope with respect to the C-radical precursor was addressed next and the direct disulfuration of unactivated

C(sp$^3$)-H bonds in various systems was investigated. Larger-ring cycloalkanes were also converted into the corresponding disulfuration products **4l**-**4n** in good yields (74–76%). Surprisingly, for the reactions of adamantane derivatives we found that better yields can be obtained in the absence of Na$_2$S$_2$O$_8$. In these cases, the adamantane derivative was used as the limiting component in combination with 1.5 equivalents of the tetrasulfide. The reaction of adamantanes bearing an electron-withdrawing group at the 1-position proceeded predominantly at the tertiary sites in 51–68% yield (see **4o**-**4r**). For 1-hydroxy- and 1-bromoadamantane, reaction occurred with complete regioselectivity, whereas the 1-adamantyl methyl ketone provided three isomers. The regioselectivities observed agree well with reported selectivities for TBADT mediated HAT processes on adamantanes and is a result of steric as well as thermodynamic effects[50,63]. Substrates bearing methyl, methylene and methine moieties react preferably at the tertiary position. Thus, the disulfides **4 s** and **4t** were isolated with good regioselectivities with little disulfuration at the terminal methyl groups. No reaction occurred at the two methylene moieties that are both deactivated by the neighboring cyano group due to polar effects. It is known

that the excited decatungstate reacts as an electrophilic H-abstractor which engages preferably with nucleophilic C–H σ-bonds[42]. Accordingly, the α- and β-positions of the ketone **3 u** are deactivated and disulfuration occurred with good regioselectivity at the terminal methyl groups (**4 u**, 50% yield, 80% regioselectivity). Note that in addition to polar effects, steric effects have to be also considered in these intermolecular HATs. The disulfuration of ketone **3 u'** gave the product **4 u'** in 42% yield with 78% regioselectivity. In analogy to the open chain ketones, the electrophilic α-position in cyclic ketones is deactivated and disulfuration proceeds at the β and γ-sites (**4 v** and **4w**, 72% and 82%). Again, with the cyclic ketone as the limiting reagent only moderate yields were obtained. As expected, a similar outcome was noted for the reaction of a cyclic sulfone (**4x**, 68%). Consistent with the hydridic nature of C–H bonds next to oxygen or sulfur atoms, functionalization of **3 y** afforded the disulfides **4 y** (62%) with excellent selectivity as single regioisomers. For the reaction with tetrahydrofuran **3z**, **4z** was obtained in 32% yield besides some unidentified byproducts. For benzylic substrates, disulfurated products **4aa** (82%) and **4bb** (86%) could be obtained in high yields at room temperature using the benzylic compound as the limiting reagent. Thus, for reactive substrates with activated C–H bonds the disulfuration works well with the alkane as the limiting reagent.

As mentioned above, S-acylated disulfides are precursors for hydropersulfides RSSH that can be used in nucleophilic substitution or metal-catalyzed cross-coupling reactions for the preparation of unsymmetrical disulfides[27,30,64]. Reported methods for the preparation of acyl alkyl disulfides generally require to synthesize complex sulfur-containing reagents or highly unstable sulfur chloride moieties[27,60,65,66]. Moreover, bisacyltetrasulfides are rather unstable and lead to acylated monsulfide products (see above). We therefore tested aldehydes as acyl radical precursors for the direct preparation of S-acylated disulfides (Fig. 5). Pleasingly, with the aldehyde as the limiting reagent using 1.5 equivalents of the tetrasulfide, functionalization of aldehydic C–H bonds worked well. For example, the aromatic 3-thienylcarbaldehyde **5a** could be transformed in a very good yield to the S-aroylated disulfide **6a** (89%). For the reaction of benzaldehyde (**5b**) and 2-naphthaldehyde (**5c**), the targeted disulfides **6b** (30%) and **6c** (46%) were formed in moderate yields besides the corresponding aroylmonosulfides (51% and 31%). However, upon shortening reaction time to 4 h, the disulfides **6b** and **6c** were obtained with excellent yields (82% and 76%, respectively). Due to product instability, decomposition to give the corresponding monosulfides is occurring at longer reaction times in these cases. Primary aliphatic aldehydes bearing a phenyl (**5d**) or a furanyl moiety (**5e**) afforded the disulfuration products **6d** and **6e** in 50% and 82% yields, respectively. Of note, aliphatic aldehydes reacted slower and 12 h reaction time was required to complete the disulfuration. As expected, primary long-chain aliphatic aldehydes worked equally well (**6f** and **6g**, 69–91% yields). The α-branched aliphatic aldehyde **5h** was converted to the disulfide **6h** (52%) and the corresponding decarbonylated dialkyl sulfide was not identified. However, with pivalaldehyde (**5i**) as a member of the tert-alkyl carbaldehyde family, the C–H functionalization reaction with the tetrasulfide **3b** provided the unsymmetrical dialkyl disulfide **6i** (89%) resulting from decarbonylation of the intermediate acyl radical. Such decarbonylation processes are often used to generate tertiary carbon radicals[67].

**Synthetic applications**. To demonstrate the utility of this methodology for late-stage functionalization of natural products and pharmaceuticals, more complex substrates were tested (Fig. 5).

**Fig. 6 Gram-scale experiments and mechanistic studies. a** Gram-scale preparation of disulfides **4a** and **6 g**. **b** Disulfuration in the presence of TEMPO and methyl acrylate.

For example, the lurasidone derivative **5j** underwent a highly regioselective radical disulfuration to afford **6j** in 72% yield with high diastereoselectivity ($dr$ = 10:1). Sclareolide, a well-studied sesquiterpenoid, reacted with tetrasulfide **3a** to provide **6k** as a single regio- and diastereoisomer (52%). The disulfuration process gave direct access to the memantine derivative **6 l** that was isolated in 71% yield and good regioselectivity. Moreover, dehydrocholic acid and indomethacin derivatives worked to give the corresponding disulfuration products **6m** and **6n** in 80% and 42% yields, respectively.

To document the practicality of the transformations, larger scale experiments were conducted at lower catalyst loading (1 mol % of **1** was used). As shown in Fig. 6a, the product **4a** was obtained in 84% yield (1.72 g) on a 5.0 mmol scale and the disulfide **6 g** was isolated in 88% yield on a 10 mmol scale, documenting the robustness of these procedures.

**Mechanistic Investigations**. The radical nature of the disulfuration was supported by control experiments. When adding 2.0 equivalents of the radical scavenger TEMPO (2,2,6,6-tetramethyl-1-piperidinyloxy), yield of the disulfuration product decreased and the TEMPO-trapping product **7** was detected by HRMS analysis (Fig. 6b). Considering the fast rate of the reaction of TEMPO with an acyl radical[27], the low amount of TEMPO trapping product is surprising. This is likely due to the fact that TEMPO gets oxidized by Na$_2$S$_2$O$_8$ under the applied conditions. We therefore ran a trapping experiment with cyclohexane in the

absence of $Na_2S_2O_8$. Indeed, TEMPO adduct **8** was isolated in 21% isolated yield along with traces of the disulfide **4a**. These results indicate that the reaction proceeds through a radical-based mechanism and that the corresponding alkyl and acyl radicals were generated under the photocatalysis conditions. Moreover, repeating the disulfuration of cyclohexane (**2a**) in the presence of 2.0 equivalents of methyl acrylate, the radical cascade product **9** was detected by HMRS analysis along with **4a**, further supporting the radical nature of the transformation (Fig. 6b).

Decatungstate photocatalysis has enabled the development of a practical radical C-H bond disulfuration strategy, that allows to access a broad range of unsymmetrical disulfides. The method works with readily prepared tetrasulfides as C-radical trapping reagents. Steered by steric and electronic effects, C-H functionalization via HAT occurs with high regioselectivity on simple and also more complex alkanes. In addition, reaction also works on aromatic and aliphatic aldehydes, where disulfuration proceeds with complete regioselectivity at the aldehydic C-H bond to afford unsymmetrical acyl disulfides. Importantly, such acylated disulfides are valuable precursors for hydropersulfides RSSH. Successful application of our strategy for late-stage C-H functionalization of more complex compounds documents its potential for the preparation of biologically important disulfide-containing compounds.

## Methods

**Representative procedures for the alkane and aldehyde disulfuration**. Alkane disulfuration. To an oven dried Schlenk tube with a magnetic stirring bar, the tetrasulfide 3 (0.3 mmol, 1.0 equiv.), the substrate 2 (3 mmol, 10 equiv.), $Na_2S_2O_8$ (0.45 mmol, 1.5 equiv.), photocatalyst TBADT (2 mol%), and 3.0 mL mixed solvent ($CH_3CN/H_2O$, $v/v$, 2/1) were added under argon atmosphere using standard Schlenk techniques at ambient temperature. After backfilling with nitrogen, the tube was placed in a photoreactor, stirred and irradiated with a Kessil 40 W 390 nm lamp for 12 h at 60 °C. The solvent was removed under reduced pressure and the crude residue was purified by silica gel column chromatography with hexane as the eluent to afford the desired product.

Aldehyde disulfuration. To an oven dried Schlenk tube with a magnetic stirring bar, the aldehyde 5 (0.3 mmol, 1.0 equiv.), the tetrasulfide 3 (0.45 mmol, 1.5 equiv.), $Na_2S_2O_8$ (0.45 mmol, 1.5 equiv.), photocatalyst TBADT (2 mol%) and 3.0 mL mixed solvent ($CH_3CN/H_2O$, $v/v$, 2/1) were added under argon atmosphere using standard Schlenk techniques at ambient temperature. After backfilling with nitrogen, the tube was placed in a photoreactor, stirred and irradiated with a Kessil 40 W 390 nm lamp for 12 h at 60 °C. The solvent was removed under reduced pressure and the crude residue was purified by silica gel column chromatography with hexane as the eluent to afford the desired product.

## Data availability

Supplementary information and chemical compound information accompany this paper at www.nature.com/ncomms. The data supporting the results of this work are included in this paper or in the Supplementary Information and are also available upon request from the corresponding author.

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

## Acknowledgements
We thank the Alexander von Humboldt Foundation (Z.J.) for supporting this work.

## Author contributions
J.J.Z. and A.S. conceived and designed the experiments. J.J.Z. performed the experiments and analyzed the data. J.J.Z. and A.S. wrote the manuscript.

## Funding

## Competing interests
The authors declare no competing interests.
