## [Peer Review File · Nature Communications]

Decatungstate-Catalyzed Radical Disulfuration Through Direct C-H Functionalization for the Preparation of Unsymmetrical DisulfidesREVIEWER COMMENTS

Reviewer #1 (Remarks to the Author):

The authors report a novel method for undirected C-H functionalization to access unsymmetrical disulfides, a class of organic compounds of interest to several areas. The protocol relies on decatungstate-catalyzed hydrogen atom transfer to generate carbon-centered radicals from C(sp³)-H and aldehydic C(sp²)-H bonds, which can then be intercepted by tetrasulfide reagents to yield disulfide products. The scope of C-H substrates and tetrasulfides is demonstrated, including several biorelevant substrates. The protocol is amenable to gram-scale setup and evidence for a carbon-centered radical mechanism is presented.

The major limitation of the protocol is the generally high stoichiometry (5-10 equiv) required for most aliphatic C-H substrates, rendering the reaction less attractive. However, due to the novelty and relevance to the field, I am willing to support publication in Nature Communications after the following issues/questions are addressed:

- 1) The authors should be clear about the yield of the protocol when using the C-H substrate as the limiting reagent, as practitioners of the field would be most interested in that for late-stage functionalization. At the very least, the cyclohexane optimization table should show the yield obtained when using 1 equiv cyclohexane. Ideally, a couple of the C(sp³)-H substrates in the scope table should also have the yield specified (NMR is fine) when they are used as the limiting reagent. Given that good yields were obtained for 1 equiv of unactivated substrates like 6j and 6k, it must be possible to obtain synthetically useful yields for some of the substrates in Figure 4, so it's strange to me that most examples are 5-10 equiv.
- 2) Do free amine substrates work in this protocol, under acidic conditions? See recent decatungstate-catalyzed examples by Britton (ACS Catal 2019, 9, 8276) and MacMillan (Nat Chem 2020, 12, 459). Given that aqueous HCl is already tolerated as cosolvent, it seems quite possible, and it would be a very valuable addition to the scope.
- 3) Do benzylic C-H substrates (e.g. toluene) work in this protocol? They typically are competent substrates in the decatungstate literature and would be a valuable addition to scope.
- 4) It is quite remarkable that benzylic disulfides (3d-3h) work in this protocol, despite having weak, easily-abstractable benzylic C-H bonds next to heteroatoms. These could compete with HAT from the desired substrate. Are good yields obtained for the products just because of the high stoichiometry of C-H substrate? Do products like 4d-4h decompose or react further if re-subjected to the reaction conditions?
- 5) The authors rule out the decatungstate catalyst turning over by reducing RSS· to RSS-, because they did not observe tBuSSH forming. However, it's possible that tBuSSH is being formed but then oxidized by the persulfate to the tetrasulfide under reaction conditions. The reaction should be reattempted with stoichiometric RSSH and no RSSSSR to see if product still forms and if RSSSSR is formed spontaneously under those conditions.
- 6) Alongside refs 40-48, some earlier work by Britton (ACIE 2014, 53, 4690) and Sorensen (Nat Commun 2015, 6, 10093) should be cited.
- 7) Footnotes for Figure 4 are a bit confusing. For instance, several substrates have both footnotes b,d, but b and d have different stoichiometries. Which is the correct one? The conditions should be made unambiguous.
- 8) In figure 5, the word "unactivated" should be removed from the panel with natural substrates, because substrates with weak, hydridic aldehyde C-H bonds like 6m and 6n are actually activated.

9) Some typos: row 98 – “entry 3” should be “entry 4”; row 146 “exited” should be “excited”; row 150 – “As for” should be “Similar to” or “Analogous to”

Reviewer #2 (Remarks to the Author):

Recommendation: Reconsider after major revisions.

Comments: Studer and Zhang described here a method to prepare unsymmetrical disulfides via direct C-H bond functionalization. It is worth mentioning that decatungstate anion ([W₁₀O₃₂]⁴⁻) was widely used in organic reactions for the generation of alkyl radicals (ref. 40-48), besides, the preparation of unsymmetrical disulfides via alkyl radical substitution with tetrasulfides was also a known process (ref. 37). This work successfully combined these two processes together and fulfilled the preparation of unsymmetrical disulfides with simple alkyl compounds or aldehydes as the substrates. After addressing the below questions, the observed reaction will be good to be reported in Nat. Commun.

1. The study of substrate scope of tetrasulfide reagents only showed the tolerance to alkyl chain, different benzyl groups, aryl group, benzoyl group and protected cysteine derivatives. What about other functional groups such as alkenyl, alkynyl, halogen substituted alkyl chain, hydroxyl and amino groups?
2. What about with cyclohexane as the substrate? Is it possible to fulfill the allyl C-H bond disulfuration process?
3. Polar effects and steric effects could account for the regioselectivity of 4u – 4w, while what about alkyl-alkyl ketones substrates with longer alkyl chains?
4. Benzyl radicals are more stable than alkyl radicals, and are easier to generate, it is necessary to fulfill the benzyl C-H bond disulfuration reaction under milder conditions.
5. The generation of aldehyde radicals should be much easier comparing with alkyl radicals, milder reaction conditions should be developed for the aldehyde C-H bond disulfuration reaction.
6. It is mentioned that diaryl, diacyl tetrasulfides and the corresponding targeted disulfides are not perfectly stable under the reaction conditions, what happened to the diaryl, diacyl tetrasulfide reagents? Would the substrates react with the metamorphic product of diaryl or diacyl tetrasulfides under the reaction conditions?
7. Some related work about the construction of unsymmetrical disulfides should be cited (Chem. Asian J. 2019, 14, 2579-2583; Org. Chem. Front., 2019,6, 2220-2225).

Reviewer #3 (Remarks to the Author):

The ms proposed by Prof. Studer and coworkers deals with the preparation of unsymmetrical disulfides promoted by a decatungstate salt and it is based on the radical addition to a tetrasulfide. The article may deserve publication in view of the importance of the products formed, but some points should be clarified.

- The first point concerns the mechanism. The involvement of carbon radicals directly formed from a hydrocarbon was demonstrated. However, previously it was demonstrated that a decatungstate salt may engage a reaction with sulfides under oxidative conditions (doi:10.2174/138527212803520092). How the Authors may exclude in this case this competitive pathway with tetrasulfides? The quenching of the excited state of the decatungstate by the tetrasulfide could be provided to exclude any interaction.
- In Table 1 it is apparent that the reaction yield is improved by adding water and by increasing the temperature up to 60 °C. Have the Authors an explanation for that? The thermal contribution to the reaction accounts for what?
- I was quite surprised to see that in the synthesis of adamantyl derivatives no persulfate is required but the Authors did not explain this unusual behaviour.
- I did not see in the text any comments on the decarbonylation occurring in the formation of

compound 6i.

Minor points: Page 8 line 142 (please correct TBATD), line 147 (ref 17a does not exist).

Reviewer #1 (Remarks to the Author):

The authors report a novel method for undirected C-H functionalization to access unsymmetrical disulfides, a class of organic compounds of interest to several areas. The protocol relies on decatungstate-catalyzed hydrogen atom transfer to generate carbon-centered radicals from C(sp³)-H and aldehydic C(sp²)-H bonds, which can then be intercepted by tetrasulfide reagents to yield disulfide products. The scope of C-H substrates and tetrasulfides is demonstrated, including several biorelevant substrates. The protocol is amenable to gram-scale setup and evidence for a carbon-centered radical mechanism is presented.

The major limitation of the protocol is the generally high stoichiometry (5-10 equiv) required for most aliphatic C-H substrates, rendering the reaction less attractive. However, due to the novelty and relevance to the field, I am willing to support publication in Nature Communications after the following issues/questions are addressed:

Reply: Thank you very much for your very nice comments and suggestions.

1) The authors should be clear about the yield of the protocol when using the C-H substrate as the limiting reagent, as practitioners of the field would be most interested in that for late-stage functionalization. At the very least, the cyclohexane optimization table should show the yield obtained when using 1 equiv cyclohexane. Ideally, a couple of the C(sp³)-H substrates in the scope table should also have the yield specified (NMR is fine) when they are used as the limiting reagent. Given that good yields were obtained for 1 equiv of unactivated substrates like 6j and 6k, it must be possible to obtain synthetically useful yields for some of the substrates in Figure 4, so it's strange to me that most examples are 5-10 equiv.

Reply: A couple of reactions in Figure 4 were repeated with the alkane as the limiting reagent (see Figure R1 below) and the revised Figure 4 in the manuscript (yields in brackets). The results obtained are discussed in the main text and the additional experiments also provided in the SI. We found that for highly challenging substrates such as cyclohexane, the yield is significantly lower if the alkane is used as the limiting reagent. However, more reactive substrates such as benzylic systems (see also Figure R3 below) or adamantanes work well. Considering late stage functionalization, luckily, most of the complex substrates have more "activated" C-H bonds as compared to cyclohexane and accordingly the radical disulfuration generally works well with complex substrate as the limiting reagent, as shown in this paper (see Figure 5).

Figure R1. The GC yields and isolated yields (for 4d-r) when the alkane is used as the limiting reagent (alkane 0.3 mmol, tetrasulfide 0.45 mmol).

2) Do free amine substrates work in this protocol, under acidic conditions? See recent decatungstate-catalyzed examples by Britton (ACS Catal 2019, 9, 8276) and MacMillan (Nat Chem 2020, 12, 459). Given that aqueous HCl is already tolerated as cosolvent, it seems quite possible, and it would be a very valuable addition to the scope.

Reply: We tried several free amine containing substrates (Figure R2). However, all of them did not work under our conditions. Also, if the amine is used as its hydrochloride salt, the reaction did unfortunately not work.

Figure R2. Various amines and amine/hydrochloride salts tested.

3) Do benzylic C-H substrates (e.g. toluene) work in this protocol? They typically are competent substrates in the decatungstate literature and would be a valuable addition to the scope.

Reply: Thank you very much for this suggestion. Benzylic C-H substrates work very well with the C-H compound as the limiting reagent (room temperature, Figure R3). The unsymmetrical disulfides 4aa and 4bb were obtained in high yields. These results were added to the revised manuscript (see Figure 4).

Figure R3. Disulfuration of benzylic C-H substrates.

4) It is quite remarkable that benzylic disulfides (3d-3h) work in this protocol, despite having weak, easily-abstractable benzylic C-H bonds next to heteroatoms. These could compete with HAT from the desired substrate. Are good yields obtained for the products just because of the high stoichiometry of C-H substrate? Do products like 4d-4h decompose or react further if re-subjected to the reaction conditions?

Reply: Although benzylic disulfides have weak C-H bonds, they are comparably stable in our conditions, and we did not observe any C-H bond functional products at that site. As requested we tested product stability and found that the products 4d-4h, 4aa and 4bb are stable if re-subjected to the reaction conditions. As mentioned above, if the cyclohexane is used as the limiting reagent yield dropped.

5) The authors rule out the decatungstate catalyst turning over by reducing RSS \cdot to RSS $^-$, because they did not observe tBuSSH forming. However, it's possible that tBuSSH is being formed but then oxidized by the persulfate to the tetrasulfide under reaction conditions. The reaction should be reattempted with

stoichiometric RSSH and no RSSSSR to see if product still forms and if RSSSSR is formed spontaneously under those conditions.

Reply: Thank you very much for your questions. We prepared cumyl hydropersulfide following a literature protocol (J. Am. Chem. Soc. 2017, 139, 6484–6493). When replacing the tetrasulfide with the cumyl hydropersulfide, the reaction did not give the desired product (Figure R4). As we know, hydropersulfides are highly reactive species and they may not be stable under our conditions. On the other hand, based on the redox potential of TBADT and persulfate ($E([W_{10}O_{32}]^{4-}/[W_{10}O_{32}]^{5-}) = -0.97 V$ vs. SCE in acetonitrile; $E(S_2O_8^{2-}/SO_4^{2-}, SO_4^{\cdot-}) = 1.1 V$ vs. SCE in acetonitrile), the electron transfer between $[W_{10}O_{32}]^{5-}$ and $S_2O_8^{2-}$ will be rather fast. Given the low concentration and poor oxidizing ability of RSS \cdot , the electron transfer between $[W_{10}O_{32}]^{5-}$ and RSS \cdot may be unfavorable.

Figure R4. The reaction of cyclohexane with the cumyl hydropersulfide.

6) Alongside refs 40-48, some earlier work by Britton (ACIE 2014, 53, 4690) and Sorensen (Nat Commun 2015, 6, 10093) should be cited.

Reply: Thank you very much for your suggestions. We have cited them in the updated manuscript (see refs 51 and 52).

7) Footnotes for Figure 4 are a bit confusing. For instance, several substrates have both footnotes b,d, but b and d have different stoichiometries. Which is the correct one? The conditions should be made unambiguous.

Reply: Thank you very much for alluding to that problem. We have clarified the conditions in the revised manuscript.

8) In figure 5, the word “unactivated” should be removed from the panel with natural substrates, because substrates with weak, hydridic aldehyde C-H bonds like 6m and 6n are actually activated.

Reply: We fully agree and accordingly have removed the word “unactivated” in the revised manuscript.

9) Some typos: row 98 – “entry 3” should be “entry 4”; row 146 “exited” should be “excited”; row 150 – “As for” should be “Similar to” or “Analogous to”

Reply: Thanks, corrected.

Reviewer #2 (Remarks to the Author):

Recommendation: Reconsider after major revisions.

Comments: Studer and Zhang described here a method to prepare unsymmetrical disulfides via direct C-H bond functionalization. It is worth mentioning that decatungstate anion ($[W_{10}O_{32}]^{4-}$) was widely used in organic reactions for the generation of alkyl radicals (ref. 40-48), besides, the preparation of unsymmetrical disulfides via alkyl radical substitution with tetrasulfides was also a known process (ref. 37). This work successfully combined these two processes together and fulfilled the preparation of unsymmetrical

disulfides with simple alkyl compounds or aldehydes as the substrates. After addressing the below questions, the observed reaction will be good to be reported in Nat. Commun.

Reply: Thank you very much for supporting our work.

1. The study of substrate scope of tetrasulfide reagents only showed the tolerance to alkyl chain, different benzyl groups, aryl group, benzoyl group and protected cysteine derivatives. What about other functional groups such as alkenyl, alkynyl, halogen substituted alkyl chain, hydroxyl and amino groups?

Reply: Considering the availability of starting materials, we prepared additional tetrasulfides (Figure R5). However, when using them as radical disulfuration reagents, only 3k' worked to give the corresponding unsymmetrical disulfide 4k' in 72% yield. This result was added to the revised manuscript (Figure 4).

Figure R5. The reaction of cyclohexane with the tetrasulfide 3k' and failed tetrasulfides.

2. What about with cyclohexane as the substrate? Is it possible to fulfill the allyl C-H bond disulfuration process?

Reply: We assume the referee was asking for cyclohexene as the substrate. We tested it under optimized conditions but the desired product was no identified.

3. Polar effects and steric effects could account for the regioselectivity of 4u – 4w, while what about alkyl-alkyl ketones substrates with longer alkyl chains?

Reply: As requested, we tested the dialkyl ketone 2u' in the reaction with 3a and the disulfuration product 4u' was obtained in 42% isolated yield with 78% regioselectivity (Figure R6). This result was added to the revised manuscript. For n-butyl methyl ketone 2u'', three regioisomeric disulfuration products 4u'' were formed as analyzed by GCMS, indicating low regioselectivity (Figure R7).

Figure R6. The reaction of the dialkyl ketone 2u' with the tetrasulfide 3a.

Figure R7. The reaction *n*-butyl methyl ketone **2u''** with the tetrasulfide **3a**.

4. Benzyl radicals are more stable than alkyl radicals, and are easier to generate, it is necessary to fulfill the benzyl C-H bond disulfuration reaction under milder conditions.

Reply: Good suggestion: we could realize the disulfuration of benzylic C-H substrates as limiting reagents at room temperature with high yields (Figure R8). These results have been added to the revised manuscript.

Figure R8. The reactions of benzylic C-H substrates with the tetrasulfide **3a** at RT.

5. The generation of aldehyde radicals should be much easier comparing with alkyl radicals, milder reaction conditions should be developed for the aldehyde C-H bond disulfuration reaction.

*Reply: The reactions of aldehydes with the tetrasulfide **3a** could be performed at room temperature, but lower yields were obtained, as shown for two cases (Figure R9). We added these results to Figure 5.*

Figure R9. The reactions of the aldehydes **5b** and **5d** with the tetrasulfide **3a** at RT.

6. It is mentioned that diaryl, diacyl tetrasulfides and the corresponding targeted disulfides are not perfectly stable under the reaction conditions, what happened to the diaryl, diacyl tetrasulfide reagents? Would the substrates react with the metamorphic product of diaryl or diacyl tetrasulfides under the reaction conditions?

Reply: When we irradiate a solution of diaryl or diacyl tetrasulfides with 390 nm light in acetonitrile, the corresponding disulfides and trisulfides were generated after 12 hours, and only a small amount of the starting tetrasulfide remained. When we used substrate 2a to react with the metamorphic products under the optimized conditions, the desired disulfuration product was not detected.

7. Some related work about the construction of unsymmetrical disulfides should be cited (Chem. Asian J. 2019, 14, 2579-2583; Org. Chem. Front., 2019,6, 2220-2225).

Reply: Thanks, we have added these citations to the revised manuscript (see refs. 16 and 17).

Reviewer #3 (Remarks to the Author):

The ms proposed by Prof. Studer and coworkers deals with the preparation of unsymmetrical disulfides promoted by a decatungstate salt and it is based on the radical addition to a tetrasulfide.

The article may deserve publication in view of the importance of the products formed, but some points should be clarified.

Reply: Thank you very much for your support and valuable suggestions.

1. The first point concerns the mechanism. The involvement of carbon radicals directly formed from a hydrocarbon was demonstrated. However, previously it was demonstrated that a decatungstate salt may engage a reaction with sulfides under oxidative conditions (doi:10.2174/138527212803520092). How the Authors may exclude in this case this competitive pathway with tetrasulfides? The quenching of the excited state of the decatungstate by the tetrasulfide could be provided to exclude any interaction.

Reply: Stern-Volmer quenching experiments showed that the tetrasulfide 3a could indeed also be oxidized with the photoexcited TBADT. As shown in Figure R10, the Stern-Volmer constant is around $1.5 \times 10^2 M^{-1}$, and we could calculate the quenching rate constant k_q ($3 \times 10^9 M^{-1}s^{-1}$) based on the lifetime of the emissive excited state of TBADT ($4.75 \times 10^{-8} s$, DOI:10.1002/ange.202104682). Moreover, referring to the literature (DOI:10.1021/jp980723t), the reaction of alkanes with the photoexcited TBADT has a comparable quenching rate constant ($1 \times 10^8 M^{-1}s^{-1}$). Although it seems reasonable that the tetrasulfide can also quench the excited photocatalyst, we are not able to draw a possible pathway how we can convert the tetrasulfide radical cation to the disulfuration product. We definitely need a C-H abstraction to generate the C-radical and the following trapping with the tetrasulfide is established and fast. Moreover, for the less activated substrates such as cyclohexane the substrate is used in excess further supporting the C-H abstraction path. For the activated substrates the H-abstraction will be faster. We therefore assume, that even if reductive quenching will happen, it is likely not a productive pathway. Moreover, we have shown the TEMPO-trapping product is formed.

Figure R10. Stern-Volmer plots for the TBADT and 3a.

2. In Table 1 it is apparent that the reaction yield is improved by adding water and by increasing the temperature up to 60 °C. Have the Authors an explanation for that? The thermal contribution to the reaction accounts for what?

Reply: Thank you very much for your question. For adding water: based on the literature (10.1016/j.apcatb.2018.09.099), the additive water mainly helps to stabilize the structure of DT anion under light illumination and increase its catalytic activity. Regarding heating: when the reaction of 2a with 3a was performed at room temperature, only 46% yield was obtained after 12 h, and a lot of metamorphic compounds of 3a (disulfide and trisulfide) were observed. When increasing the reaction temperature to 60 °C, the product 4a could be obtained in 86% yield. As shown above, the metamorphic products are not converted to the targeted disulfides. Thus at higher temperature the desired reaction is more accelerated than the decomposition of the tetrasulfide to the corresponding metamorphic sulfides.

3. I was quite surprised to see that in the synthesis of adamantyl derivatives no persulfate is required but the Authors did not explain this unusual behavior.

Reply: This is an experimental finding. Honestly, we do not understand and are therefore not able to comment that experimental finding. Rather than speculating we will not comment in the paper.

4. I did not see in the text any comments on the decarbonylation occurring in the formation of compound 6i.

Reply: We have added the sentence to the revised manuscript “Such decarbonylation processes are often used to generate tertiary carbon radicals” and also added a reference regarding the decarbonylation reaction (see ref. 67).

5. Minor points: Page 8 line 142 (please correct TBATD), line 147 (ref 17a does not exist).

Reply: Thanks, corrected.

REVIEWERS' COMMENTS

Reviewer #1 (Remarks to the Author):

The authors have adequately addressed the requested revisions. I support publication of this work in Nature Communications.

Reviewer #2 (Remarks to the Author):

Recommendation: Accepted

In my opinion, the authors have properly improved the work in this revision of the manuscript. This work should be published in Nature Communication.

Reviewer #3 (Remarks to the Author):

I had a look to the Author's comments regarding my issues raised in the ms. I only suggest (for the sake of completeness) to quote in the ms that the tetrasulfide 3a could indeed also be oxidized with the photoexcited TBADT (the Stern-Volmer plots should be included in the SI and commented). Apart this, the ms could be published in the present form.

Reviewer #1 (Remarks to the Author):

The authors have adequately addressed the requested revisions. I support publication of this work in Nature Communications.

Reply: Thank you very much for your support!

Reviewer #2 (Remarks to the Author):

Recommendation: Accepted

In my opinion, the authors have properly improved the work in this revision of the manuscript. This work should be published in Nature Communication.

Reply: Thank you very much for your support!

Reviewer #3 (Remarks to the Author):

I had a look to the Author's comments regarding my issues raised in the ms. I only suggest (for the sake of completeness) to quote in the ms that the tetrasulfide 3a could indeed also be oxidized with the photoexcited TBADT (the Stern-Volmer plots should be included in the SI and commented).

Apart this, the ms could be published in the present form.

Reply: Thank you very much for your support! We have added the following comment to the revised manuscript and added the data to the SI. "The initial C-radical generation via intermolecular HAT to the photoexcited TBADT is very well established, as already discussed above.⁴²⁻⁵² Although Stern-Volmer quenching experiments showed that the tetrasulfide 3a could also be oxidized with the photoexcited TBADT (see supplementary material), a reasonable reaction path to the product disulfide following this activation mode could not be drawn."